# How common is patient and public involvement (PPI)? Cross-sectional analysis of frequency of PPI reporting in health research papers and associations with methods, funding sources and other factors

Iain Lang 🆔 , Angela King, Georgia Jenkins, Kate Boddy 🆔 , Zohrah Khan, Kristin Liabo

University of Exeter Medical School, University of Exeter, Exeter, UK

**Correspondence to**
Dr Iain Lang;
i.lang@exeter.ac.uk

## ABSTRACT

**Objectives** Patient and public involvement (PPI) in health research is required by some funders and publications but we know little about how common it is. In this study we estimated the frequency of PPI inclusion in health research papers and analysed how it varied in relation to research topics, methods, funding sources and geographical regions.

**Design** Cross-sectional.

**Methods** Our sample consisted of 3000 research papers published in 2020 in a general health-research journal (*BMJ Open*) that requires a statement on whether studies included PPI. We classified each paper as 'included PPI' or 'did not include PPI' and analysed the association of this classification with location (country or region of the world), methods used, research topic (journal section) and funding source. We used adjusted regression models to estimate incident rate ratios of PPI inclusion in relation to these differences.

**Results** 618 (20.6%) of the papers in our sample included PPI. The proportion of papers including PPI varied in relation to location (from 44.5% (95% CI 40.8% to 48.5%) in papers from the UK to 3.4% (95% CI 1.5% to 5.3%) in papers from China), method (from 38.6% (95% CI 27.1% to 50.1%) of mixed-methods papers to 5.3% (95% CI −1.9% to 12.5%) of simulation papers), topic (from 36.9% (95% CI 29.1% to 44.7%) of papers on mental health to 3.4% (95% CI −1.3% to 8.2%) of papers on medical education and training, and funding source (from 57.2% (95% CI 51.8% to 62.6%) in papers that received funding from the UK's National Institute for Health Research to 3.4% (95% CI 0.7% to 6.0%) in papers that received funding from a Chinese state funder).

**Conclusions** Most research papers in our sample did not include PPI and PPI inclusion varied widely in relation to location, methods, topic and funding source.

## BACKGROUND

The involvement and engagement of patients and members of the public in the conduct

## STRENGTHS AND LIMITATIONS OF THIS STUDY

⇒ There has been little previous research on how often patient and public involvement (PPI) is included in health research papers and our sample (of 3000 papers) is much larger than that of previous studies.

⇒ As far as we are aware, this is the first study to analyse associations between PPI inclusion and other aspects of research papers such as topic and methods.

⇒ Our sample came from a single journal, but it is a general health-research and medical-research journal that includes a broad range of topics and methods and requires authors to provide a statement on PPI inclusion.

⇒ We relied on published statements provided in research papers and cannot assess their accuracy.

of health research has been endorsed and promoted by a range of government and other funders.[1] The roots of this lie in the WHO's Alma Alta Declaration,[2] the product of a major international conference on public health that has had a lasting impact on health and healthcare, which stated that people have a 'right and duty to participate individually and collectively in the planning and implementation of their healthcare'. Those who promote patient and public involvement (PPI) in health research often argue for it based on one or both of two reasons, one normative and one practical.[3] The normative reason is that PPI is seen as making research more democratic and egalitarian and to move activity towards research done *with* or *by* people rather than *on* or *about* them.[4] The practical reason is that PPI is seen as benefiting not only the process of research, for example, by improving recruitment and



retention of participants in clinical trials,[5] but also the quality of research and the impact of research findings.[6–8] As a means of implementing PPI in health research, some research funders now recommend or stipulate PPI in the work they fund. This is done by some state funders, like the UK's National Institute for Health Research[9] and the Australian National Health and Medical Research Council,[10] as well as by charitable funders such as the UK's Alzheimer's Society[11] and McPin Foundation.[12]

However, reviews of PPI have found low levels of involvement and poor-quality reporting.[13–17] Even in areas where we might expect high levels of patient involvement, such as trials of patient-oriented research or the development of patient-reported outcome measures, PPI is often poorly reported or absent.[18 19] Research into PPI has identified concerns among researchers grounded in perceptions of the costs involved,[20] uncertainties about how to do it[21] or report it,[22] and difficulties related to issues such as tokenism, box-ticking and the role of PPI as a form of governance.[23–27]

To date, research on the inclusion of PPI has been almost entirely qualitative and we lack information on how widespread PPI is within health research and in what circumstances researchers are more or less likely to include it: information that is important if we are to understand and promote PPI. Our aim in this study was to provide some of this information. We used data on PPI in papers published in BMJ Open, an open-access journal in which PPI reporting is obligatory and authors are required to include a statement about PPI even if they did not include PPI in their study. We looked at all the PPI statements in all research papers published in BMJ Open in a 12-month period. We analysed, first, how common PPI inclusion was and, second, whether and to what extent PPI varied in relation to location, method, topic and funding source.

## METHODS
### Data collection
We used ProQuest, an online bibliographic database (www.proquest.com), to identify all original research papers published in BMJ Open during the 12 months from 1 January 2020 to 31 December 2020 (n=3000), the most recent complete year for which data were available when we started data collection. The data for our study were the full texts of papers plus accompanying metadata (information on each paper detailing date of publication, document IDs, journal section and so on). We used a custom-written Perl[28] script to read the data file, extract relevant information and assemble our analysis data set.

For each paper, we extracted:
► *First author name*, for ease of reference and reporting.
► The *URL* that linked directly to the online version of the paper using its digital object identifier and which also constituted a unique identifier for each paper.
► *Patient and public involvement statement*: full text of the statement made in the paper (as part of the

main text) or following the paper (alongside the acknowledgements).
► *Location* (country or region) from which the paper came, based on the country of the primary affiliation of the first author. Many countries were associated with only a small number of papers and for this reason, as well as for ease and clarity of reporting, we bundled all countries with fewer than 100 associated papers into regions. This left us with seven individual countries and six regions.
► *Method,* based on our categorisation of words in the title, from which we identified 18 terms that allowed us to assign each paper to at least one category (systematic review, protocol, trial, etc) in relation to methods. Some papers had more than one method associated with them because they contained multiple terms, for example, both protocol and trial.
► *Research Topic,* based on the section of the journal to which the paper was assigned. Most of these relate to substantive topics (public health, cardiovascular medicine, nursing, etc) and some relate to methods (epidemiology, qualitative research, health economics, etc) but each paper is assigned to only one section. The papers in our sample fell into 61 sections but many of these had only small numbers of papers in them. For reporting purposes, below, we provide details of research topics from the 19 sections containing at least 50 papers.
► *Funding source*, extracted from the funding section of each paper. We identified all funders associated with each paper but for reporting purposes we only provide details of those with at least 25 papers related to them: there were 12 of these. These more common funders were associated with approximately one-third of the papers in our sample.

We looked at four *exposure variables* based on this extracted information: location, method, research topic and funding source. Our aim was to understand whether papers that differed in relation to these categories (differences in location, differences in methods and so on) were also likely to differ in terms of including PPI.

We based our *outcome variable* on our interpretation of the PPI statement and categorised each paper as either 'included PPI' or 'did not include PPI'. We used the Perl script to find phrases commonly used in PPI statements, such as 'no patients involved', 'patients and public were not involved' and 'patients and/or the public were not invited'. Based on this, the script assigned 1797 of the 3000 papers to the 'did not include PPI' group. This assignation helped us reach our final decision on the group in which we should put each paper but did not constitute our decision: we read through each statement to check that we agreed with the categorisation.

Deciding what does and does not constitute PPI is conceptually difficult, especially when only limited information—short statements provided by authors—is available. We classed papers as 'included PPI' if any form of involvement by one or more non-professional

researchers, whether as individuals or in the form of a group, was claimed at any stage of the study. We classed as 'did not include PPI' papers in which:

► PPI was explicitly omitted (eg, 'Patients and the public were not involved in this research study').[29]
► Patient input was taken from previous literature or previous studies (eg, 'Previous studies about patient experiences in open-label placebo (OLP) and placebo trials of irritable bowel syndrome were consulted for this study to include patient experiences').[30]
► PPI-type input was provided by experts whose primary role was not to provide knowledge from lived experience or public oversight of the study (eg, 'The indication, research questions and study endpoint outcome measures were selected based on the authors' expert understanding in the care of affected patients, their needs and therapy preferences, without direct communication of the study design to patients').[31]

One of the authors (IL) took responsibility for reading all the PPI statements, correcting the script-based assignments where they were wrong, and making a final decision on the categorisation of each paper. IL and two other authors (KB and KL) did initial calibration for this decision-making in concert: all three of us discussed several borderline cases and arrived at a consensus on how to classify these. In 130 of the papers in our data set we could not find a statement or any other information about PPI and we coded these as not including PPI.

### Analysis

We analysed our data using Stata/SE V.17.0 and Microsoft Excel. For each exposure category (location, method, research topic, funder) we calculated descriptive statistics relating to the number of paper assigned to each category and the mean level of PPI in each. For example, in relation to methods, did papers that reported trials, evaluations or systematic reviews differ in terms of how likely they were to include PPI? What about papers that concerned surgery compared with those about anaesthetics or general practice? For the groupings within each category we estimated, with 95% CIs, the proportion of papers in which the authors reported the inclusion of PPI.

In the regression models of which we report the results below, we used Stata to estimate the relative risk and CIs of having PPI by means of generalised linear models with a modified Poisson approach and robust error variances.[32] To account for potential confounding, we adjusted our models for location, method and topic for each of the other exposure variables (location/method/topic/funding source). When the outcome was funding source we adjusted for method and topic but not for location, because location and funding source were often overlapping. For example, of the 178 papers receiving funding from Chinese state funders, 174 (97.8%) came from China—see additional analyses below.

The highest and lowest levels of PPI inclusion were associated, respectively, with papers receiving funding from two national government funders, the UK's National

Institute for Health Research (NIHR) and Chinese state funders. To provide insight into the relationships with PPI inclusion of being based in a specific country and of being supported by the state funder in that country, we conducted additional analyses to look at the differences in PPI inclusion levels in UK papers that were and that were not funded by NIHR, and in Chinese papers that were and were not funded by Chinese state funders.

### Ethics approval

Since this study used publicly available data and involved neither human nor animal subjects, we did not submit it for, and it has not received, formal ethics approval.

### PPI

One of our coauthors, AK, is not a professional researcher but has experience of being involved in research as a patient and carer. She is a member of the Peninsula Patient Engagement Group (https://arc-swp.nihr.ac.uk/ppie/groups-we-work-with/penpeg), has been a patient and carer for most of her life, and has remained active in both patient advocacy and PPI for over 25 years. She joined the study after inception and was involved in data analysis and interpretation and in writing the manuscript. From the perspective of the professional coauthors, AK's most important contribution has been to challenge us on how we have described PPI throughout the paper. She drew our attention to issues around language and power in an early draft and because of this we changed the way in which we described PPI throughout the paper.

### RESULTS

Of the 3000 papers in our sample, we identified 618 (20.6%, 95% CI 19.1% to 22.0%) as having included PPI. The tables and figures below show the distribution of these papers in relation to the four types of difference we examined: location (table 1), method (table 2), research topic (table 3) and funder (table 4). Each table shows the unadjusted estimates of mean PPI in each category, ordered from greatest to lowest:

► In relation to *location* (table 1), the likelihood of PPI inclusion ranged from 44.5% (95% CI 40.5% to 48.5%) in the UK to 3.4% (95% CI 1.5% to 5.3%) in China.
► In relation to *method* (table 2), the likelihood of PPI inclusion ranged from 38.6% (95% CI 27.1% to 50.1%) in mixed-methods papers to 5.3% (95% CI −1.9% to 12.5%) in simulation papers.
► In relation to *research topic* (table 3), the likelihood of PPI inclusion ranged from 36.9% (95% CI 29.1% to 44.7%) in papers on mental health to 3.4% (95% CI −1.3% to 8.2%) of papers on medical education and training.
► In relation to *funding source* (table 4), the likelihood of PPI inclusion ranged from 57.2% (95% CI 51.8% to 62.6%) in papers that received funding from NIHR to



**Table 1** Number of papers published in BMJ Open in 2020 by location, and proportion of these with PPI

| Location | n | % of total | Mean % PPI (95% CI) |
|---|---|---|---|
| UK | 591 | 19.7 | 44.5 (40.5 to 48.5) |
| Canada | 195 | 6.5 | 30.8 (24.3 to 37.3) |
| Australia/New Zealand | 269 | 9.0 | 21.9 (17.0 to 26.9) |
| France | 118 | 3.9 | 21.2 (13.8 to 28.6) |
| Netherlands | 120 | 4.0 | 20.8 (13.5 to 28.1) |
| Germany | 101 | 3.4 | 16.8 (9.5 to 24.2) |
| USA | 227 | 7.6 | 16.3 (11.5 to 21.1) |
| Europe (other) | 501 | 16.7 | 15.5 (12.4 to 18.7) |
| Central and South America | 59 | 2.0 | 11.3 (3.9 to 18.7) |
| Central and South Asia | 72 | 2.4 | 10.2 (2.4 to 18.0) |
| Africa and West Asia | 163 | 5.4 | 8.0 (3.8 to 12.1) |
| East and Southeast Asia | 232 | 7.7 | 6.4 (3.3 to 9.6) |
| China | 352 | 11.7 | 3.4 (1.5 to 5.3) |
| | 3000 | 100.0 | 20.6 (19.1 to 22.0) |

Note that location here means each country with ≥100 papers and otherwise means region.
PPI, patient and public involvement.

**Table 2** Number of papers published in BMJ Open in 2020 by method used, and proportion of these with PPI

| Method | n | % of total | Mean % PPI (95% CI) |
|---|---|---|---|
| Mixed methods | 70 | 2.3 | 38.6 (27.1 to 50.1) |
| Implementation | 65 | 2.2 | 35.4 (23.7 to 47.1) |
| Trial | 551 | 18.4 | 34.8 (30.9 to 38.8) |
| Qualitative | 260 | 8.7 | 34.2 (28.4 to 40.0) |
| Evaluation | 115 | 3.8 | 33.0 (24.4 to 41.7) |
| Pilot study | 42 | 1.4 | 31.0 (16.8 to 45.1) |
| Development | 118 | 3.9 | 29.7 (21.3 to 37.9) |
| Protocol | 1181 | 39.3 | 28.9 (26.3 to 31.5) |
| Health economics | 91 | 3.0 | 25.3 (16.3 to 34.3) |
| Survey | 158 | 5.3 | 17.7 (11.7 to 23.7) |
| Observational | 220 | 7.3 | 15.9 (11.1 to 20.8) |
| Cohort | 637 | 21.2 | 14.6 (11.9 to 17.3) |
| Scoping | 125 | 4.2 | 13.6 (7.6 to 19.6) |
| Time series | 25 | 0.8 | 12.0 (−1.0 to 25.0) |
| Systematic review | 376 | 12.5 | 9.6 (6.6 to 12.6) |
| Cross-sectional | 464 | 15.5 | 9.3 (6.6 to 11.9) |
| Meta-analysis | 265 | 8.9 | 8.3 (5.0 to 11.6) |
| Simulation | 38 | 1.3 | 5.3 (−1.9 to 12.5) |

Note that total n and % of papers here each sum to more than the total number of papers or to >100% because some have more than one designation—for example, both protocol and trial.
PPI, patient and public involvement.

3.4% (95% CI 0.7% to 6.0%) in papers that received funding from a Chinese state funder.

Papers that did not receive funding from any of the major funders (that is, those included in table 4) (n=2028) had a mean level of PPI inclusion of 14.9% (95% CI 13.4% to 16.5%), which is statistically significantly lower than the mean level of PPI inclusion of the whole data set.

Figures 1–4 show the relative risks and CIs of each category estimated using an adjusted generalised linear model as described above for location, method, research topic and funding source, respectively.

### Additional analyses

Of the 320 papers whose authors reported receiving funding from the UK's NIHR, 272 (85.0%) came from the UK. Those 272 papers represent 46.0% of the 591 UK papers in our sample, and 59.9% (95% CI 54.1% to 65.7%) of them included PPI. Of the 319 (54.0%) of UK papers whose authors did not report receiving NIHR funding, 31.3% (95% CI 26.2% to 36.5%) included PPI. For UK papers, receipt of NIHR funding was associated with increased PPI inclusion.

Of the 178 papers whose authors reported receiving funding from Chinese state funders, 174 (97.8%) came from China. Those 174 papers represent 49.3% of the 352 Chinese papers in our sample, and 3.4% (95% CI 0.7% to 6.2%) of them included PPI. Of the 178 (50.6%)

of Chinese papers whose authors did not report receiving Chinese state funding, 3.4% (95% CI 0.7% to 6.0%) included PPI. For Chinese papers, receipt of Chinese state funding was not associated with any difference in PPI inclusion.

### DISCUSSION

We had two aims in this study: to assess how common PPI is in a sample of published health research papers, and to measure how much variation there is in PPI inclusion according to known differences between the papers. We addressed these aims by looking at the PPI statements included in 3000 papers published in a general health and medical research journal where authors are required to include a statement about PPI, BMJ Open. In relation to our first aim, we found that approximately one in five (20.6%) of the papers in our sample reported including PPI. In relation to the second, we found marked variations in PPI inclusion across the categories we looked at. For example, papers from the UK were more than 10 times likelier to include PPI than those from China, and there were also marked differences in relation to each of method, topic and funding source.

These large differences may indicate where there is potential to increase PPI inclusion in research. For

Table 3  Number of papers published in BMJ Open in 2020 in each research topic with ≥50 papers, and proportion of these with PPI

| Research topic | n | % of total | Mean % PPI (95% CI) |
|---|---|---|---|
| Mental health | 149 | 5.0 | 36.9 (29.1 to 44.7) |
| Qualitative research | 51 | 1.7 | 35.3 (22.0 to 48.5) |
| Health services research | 168 | 5.6 | 31.0 (23.9 to 38.0) |
| Neurology | 82 | 2.7 | 29.3 (19.4 to 39.2) |
| Surgery | 102 | 3.4 | 28.4 (19.6 to 37.2) |
| Rehabilitation medicine | 65 | 2.2 | 26.2 (15.4 to 36.9) |
| Oncology | 88 | 2.9 | 25.0 (15.9 to 34.1) |
| Cardiovascular medicine | 120 | 4.0 | 24.2 (16.5 to 31.9) |
| Paediatrics | 91 | 3.0 | 23.1 (14.4 to 31.8) |
| Diabetes and endocrinology | 85 | 2.8 | 22.4 (13.4 to 31.3) |
| Obstetrics and gynaecology | 98 | 3.3 | 21.4 (13.3 to 29.6) |
| Emergency medicine | 66 | 2.2 | 21.2 (11.3 to 31.2) |
| General practice/ family practice | 94 | 3.1 | 20.2 (12.0 to 28.4) |
| Public health | 366 | 12.2 | 16.1 (12.3 to 19.9) |
| Infectious diseases | 79 | 2.6 | 15.2 (7.2 to 23.2) |
| Global health | 97 | 3.2 | 13.4 (6.6 to 20.2) |
| Health economics | 61 | 2.0 | 6.6 (0.3 to 12.8) |
| Epidemiology | 241 | 8.0 | 6.2 (3.2 to 9.3) |
| Medical education and training | 58 | 1.9 | 3.4 (−1.3 to 8.2) |
| | 1261 | 71.8 | |

PPI, patient and public involvement.

Table 4  Number of papers published in BMJ Open in 2020 by funding source for funders named in ≥25 papers, and proportion of these with PPI

| Funder | n | % of total | Mean % PPI (95% CI) |
|---|---|---|---|
| NIHR | 320 | 10.7 | 57.2 (51.8 to 62.6) |
| CIHR | 90 | 3.0 | 41.1 (30.9 to 51.3) |
| ZonMw | 25 | 0.8 | 40.0 (20.4 to 59.6) |
| MRC | 58 | 1.9 | 39.7 (27.0 to 52.4) |
| ESRC | 41 | 1.4 | 29.3 (15.2 to 43.4) |
| Wellcome Trust | 103 | 3.4 | 28.2 (19.4 to 36.9) |
| NHMRC | 111 | 3.7 | 27.9 (19.5 to 36.3) |
| NIH | 81 | 2.7 | 27.2 (17.4 to 36.9) |
| EU | 69 | 2.3 | 23.2 (13.2 to 33.2) |
| Carlos III | 30 | 1.0 | 13.3 (1.0 to 25.7) |
| Gates Foundation | 33 | 1.1 | 12.1 (0.8 to 23.4) |
| Chinese | 178 | 5.9 | 3.4 (0.7 to 6.0) |

Carlos III, Instituto de Salud Carlos III (Spain); Chinese, Key Research or other Chinese state funder, for example, National Science Foundation of China; CIHR, Canadian Institutes of Health Research; ESRC, Economic and Social Research Council (UK); EU, European Union funding; Gates, The Bill and Melinda Gates Foundation (USA); MRC, Medical Research Council (UK); NHMRC, National Health and Medical Research Council (Australia); NIH, US National Institutes of Health; NIHR, National Institute of Health Research (UK); PPI, patient and public involvement; Wellcome, The Wellcome Trust (UK); ZonMw, The Netherlands Organisation for Health Research and Development.

example, funders wishing to increase PPI inclusion might target support and promotion to researchers working in fields with low levels of inclusion. The funding source we found to be associated with the highest level of PPI inclusion was the UK's NIHR, which has for several years included in its application process a section in which applicants may either apply for funding to include PPI or justify why they are not including it. We note, at the same time, that NIHR lacks a single, unequivocal policy statement about PPI.[33]

Previous work has reviewed PPI reporting in specific topics such as dementia,[14 34] patient-oriented-research trials[18] and surgical trials.[15] Although the authors of these studies did not attempt to measure the proportion of published papers involving PPI, as we have done

here, they uniformly identified few studies reporting PPI within the large literatures in which they were interested; a study of clinical trials published in nursing journals failed to find any evidence of PPI.[35] We know of only one recent study on the prevalence of PPI in general medical research, which, in a smaller sample, found that 11% of papers reported PPI inclusion.[36] We are unaware of any

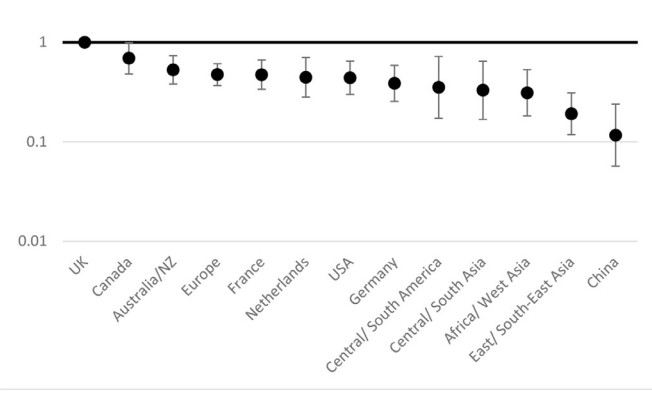

Figure 1  Relative risk of patient and public involvement (incidence-rate ratio) in studies by location compared to UK, with 95% CIs, adjusted by method, topic and funder. Incidence-rate ratios (y-axis) are on a logarithmic scale (base 10). Location here means each country with ≥100 papers and otherwise means region.

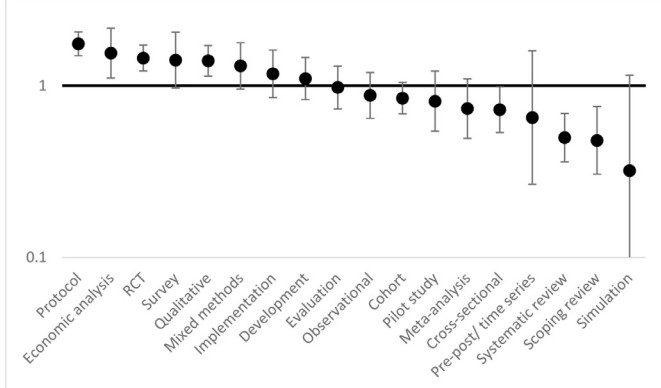

**Figure 2** Relative risk (incidence-rate ratio) of patient and public involvement in studies by method, with 95% CIs, adjusted by location, topic and funder. Incidence-rate ratios (y-axis) are on a logarithmic scale (base 10). RCT, randomised controlled trial.

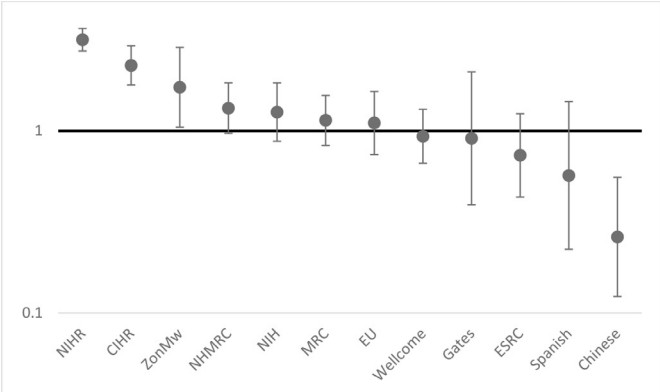

**Figure 4** Relative risk (incidence-rate ratio) of PPI in studies by funding source compared with mean level, with 95% CIs, adjusted by method and topic. Carlos III, Instituto de Salud Carlos III (Spain); Chinese, Key Research or other Chinese state funder, for example, National Science Foundation of China; CIHR, Canadian Institutes of Health Research; ESRC, Economic and Social Research Council (UK); EU, European Union funding; Gates, The Bill and Melinda Gates Foundation (USA); MRC, Medical Research Council; NIH, US National Institutes of Health; NIHR, National Institute of Health Research (UK); NHMRC, National Health and Medical Research Council (Australia); Wellcome, The Wellcome Trust (UK); ZonMw, The Netherlands Organisation for Health Research and Development. Incidence-rate ratios (y-axis) are on a logarithmic scale (base 10).

other studies on the prevalence of PPI inclusion or of factors associated with it.

Our study has some noteworthy limitations. The first of these relates to PPI statements. We took authors' PPI statements at face value: if the statement said that PPI took place then we took this to be the case, even though there is evidence that what researchers say they will do and what they actually do, in relation to PPI, varies.[37] Authors may have exaggerated, underplayed or otherwise misreported the extent of PPI in their studies and the findings we have reported do not allow us to say anything about the quality of the PPI in the 'included PPI' studies. We had no way of telling whether PPI statements reflect the experience of involvement of the individuals or groups providing the PPI and note that statements about participation and involvement are often performative[38] and always political.[39] However, we note the conclusion of a previous study: that apparently low levels of PPI inclusion are not simply due to under-reporting.[36]

A second, related, limitation is that PPI may be understood differently in different places and, as a result,

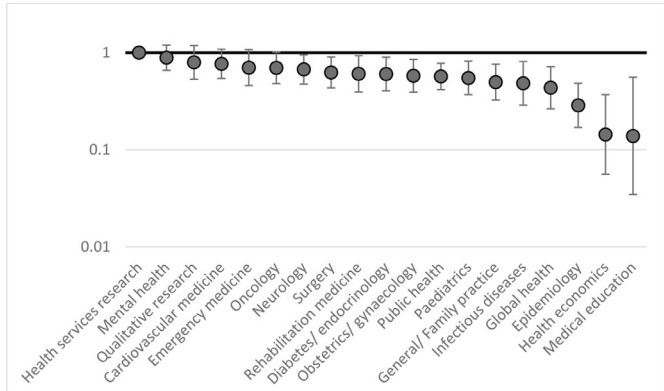

**Figure 3** Relative risk (incidence-rate ratio) of patient and public involvement in studies by topic compared to health services research, with 95% CIs, adjusted by location, method and funder. Incidence-rate ratios (y-axis) are on a logarithmic scale (base 10).

reported differently. We found PPI is more common in health research from the UK than elsewhere. The development and institutionalisation of PPI in the UK can be seen as the outcome of specific national sociopolitical events and trends. As a result, the term 'patient and public involvement' has resonances in UK health research that may be different or absent elsewhere. As a group of authors, we are most familiar with policies and expectations around PPI in the country in which we are based, the UK, so we cannot comment in detail on the ways PPI may be viewed or implemented elsewhere. In other settings there may be other ways of addressing the concerns that PPI is seen as addressing (eg, through working with patient-interest organisations rather than with individual patient representatives) and a focus on PPI may overlook these. A recent systematic review of attitudes and approaches to PPI in Europe[40] found a convergence of understandings in some European nations but lack of institutional support in many places. Although not focusing on research, Zhu[41] describes how patient involvement and participation in Chinese hospitals is influenced by factors including level of civil society engagement, moves towards marketisation and individuals' cultural resources. Possibilities for and meanings of PPI may vary widely in relation to both national and subnational differences and the differences may be subtle: Fang[42] has written about the challenges he, a Chinese immigrant to the USA, faces when preparing scholarly works in English, which has an entirely different set of norms and expectations related to how scholars should write and argue.

A third limitation relates to our sample. We used a purposeful sample of papers from a single journal, BMJ Open—if this were a clinical study we might call it a case series. We used BMJ Open because, as far as we know, it is the only journal that (a) has a written policy requiring papers to include a PPI statement *and* (b) is open access *and* (c) is not discipline- or method-specific and publishes papers across a range of health-related topics and range of methods. Using only a single journal also simplified our analysis, since we did not have to consider or attempt to account for variations across journals in terms of editorial policy or other factors. However, peer-reviewed journals are, by definition, selective in the papers in they publish. BMJ Open publishes many study protocols and many papers by authors based in the UK. The journal's policy regarding PPI may encourage authors to submit papers that include PPI and discourage them from submitting those that lack it. These characteristics may influence the topics and methods submitted to the journal as well as actions concerning PPI.

We believe there is scope for further research on the issues we have begun to explore here. This might include work on the quality of PPI reporting, perhaps based on established criteria; replication of our analyses using a sample of papers taken from elsewhere; and qualitative analysis of the statements made both by authors who report PPI and those who report none. Our dichotomisation of statements into 'included PPI' or 'did not include PPI' ignores any issues of quality or extent of involvement and this topic merits investigation. Our analysis has looked only at associations between PPI involvement and broad categories such as country of origin or funding body and not micro-level differences such as attitudes, policies or beliefs at the level of research organisations, research groups or individual researchers. We will need different methods to study such influences on how and why PPI in health research occurs—and why it does not.

**Contributors** Concept and design, statistical analysis and drafting of the manuscript: IL. Acquisition, analysis or interpretation of the data, critical revision of the manuscript for important intellectual content, approval of final version and agreement to be accountable for all aspects of the work: IL, AK, GJ, KB, ZK, KL. IL acts as the guarantor for this study and as such accepts full responsibility for the conduct of the study, had access to the data, and controlled the decision to publish.

**Funding** This report is independent research supported by the UK National Institute for Health Research Applied Research Collaboration South West Peninsula (Grant Reference Number: NIHR200167). The views expressed in this publication are those of the author(s) and not necessarily those of the National Institute for Health Research or the UK Department of Health and Social Care.

**Competing interests** None declared.

**Patient and public involvement** Patients and/or the public were involved in the design, or conduct, or reporting, or dissemination plans of this research. Refer to the Methods section for further details.

**Patient consent for publication** Not applicable.

**Ethics approval** Not applicable.

**Provenance and peer review** Not commissioned; externally peer reviewed.

**Data availability statement** Data sharing not applicable as no data sets generated and/or analysed for this study. The data we used are drawn from the texts and metadata of articles published in an open-access journal, BMJ Open. The data are publicly available online.

**ORCID iDs**
Iain Lang http://orcid.org/0000-0002-8473-2350
Kate Boddy http://orcid.org/0000-0001-9135-5488

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
