## [Reviewer comments · BMJ Open]

ARTICLE DETAILS

TITLE (PROVISIONAL)	How common is Patient and Public Involvement (PPI)? Cross-sectional analysis of frequency of PPI reporting in health research papers and associations with methods, funding sources, and other factors
AUTHORS	Lang, Iain; King, Angela; Jenkins, Georgia; Boddy, Kate; Khan, Zohrah; Liabo, Kristin

VERSION 1 – REVIEW

REVIEWER	Collins, Simon HIV i-Base
REVIEW RETURNED	13-Apr-2022

GENERAL COMMENTS	Thank you for the chance to review this cross-sectional study that reports patient and public involvement (PPI) in roughly 20% of the research papers published in BMJ Open during 2020 (618/3000). Also for the regional breakdown by country and by type of study, The paper is well written and the results provide a snapshot indication that PPI is under used, with less than half the studies from the UK (the country with highest report at 46%) reporting any involvement. The results are important for showing how few studies currently use PPI globally. even in countries where this concept is broadly understand, such as the UK. The following comments and mainly minor suggestions. Because the professional authors in the writing group are all from a single centre, this perhaps misses an opportunity to include wider regional diversity, which might have helped a study reporting on international differences. Page 5 - lines 3-12 How did the group recruit the individual PPI experience? Was this a friend or colleague who also happened to have been a patient/carer? It would help to know why the group did not include someone who formally represented a patient or public group - and who had this broader experience. It is good that the PPI contribution is recognised at co-author level. However, this person was only engaged after the study had been designed. It would have made more sense for any study on PPI to
--

	really have started with PPI engagement from first principles. This aspect of the study could perhaps be included as a study limitation. The groups own approach could perhaps be used in the discussion. Discussion General - Do the group think that there are studies where PPI engagement is not appropriate? It would be useful to give examples Page 12 - line 10 onwards This paragraph is sufficiently distinct from the first limitation to be referred to as a second limitation. The UK focus is a very important limitation given the international aspects of the findings. Perhaps the discussion could address likely large cultural and regional differences in the PPI concept, for example in general between high vs low income (in terms of level of support for research). Also, in UK/Canada/Australia and USA vs China/SE Asia etc (in terms of English language). It might be helpful to include a table that summarises country/regional policies on PPI and the level of requirement in regional funding agencies. As this limitation could undermine the differences reported by region, the discussion could include a counter argument against this. Otherwise, the results currently read like a UK vs China comparison when the basis and understanding on PPI are likely to be very different in each setting (invalidating such a simple comparison). Page 25, line 25 If point above is accepted, this should become the third limitation. The use of one journal might not be a limitation, because it standardises one aspect of the study question. Perhaps comment in detail on why inclusion in countries with high recommendations to include PPI are still so limited. If the authors have limited information about policies in other regions and countries, this should be listed as a limitation. Otherwise, the paper is reporting on an issue that they know is established in their own country but commenting on research in very different settings. Did the group consider the inclusion of a community rep as a co-author as an indication of the level of PPI - and for formally acknowledging this level of engagement? Please could the discussion include suggestions for how the present level of PPI could be increased - especially when funders not only strongly encourage this but are willing to cover costs. What about the role of publishers in influencing PPI (as a criterion
--	--

	for publication). Will future projects include contacting research studies without PPI to understand reasons for exclusion?
--	--

REVIEWER	Talbot, Amelia University of Oxford, Nuffield Department of Primary Health Care Sciences
REVIEW RETURNED	18-Apr-2022

GENERAL COMMENTS	This research article presents a cross-sectional analysis of 3000 papers in the BMJ open published in 2020 to conclude on frequency of usage and reporting of patient and public engagement (PPI). I thank the BMJ Open and the authors for allowing me to review this insightful and generally well-written article which I hope will get funders, journals, and researcher to reflect on the importance of conducting and reporting PPI. This paper I feel is appropriately submitted to the BMJ Open given that it samples articles for this journal. For full transparency, I am a qualitative researcher with limited experience in quantitative methods. So, my comments on your results are limited. I normally do not review non-qualitative papers. However, I am currently doing a massive research project on being a patient researcher (I research the condition I live with). As such, I have a vast array of knowledge on PPI and feel that my comments may be of use to you. I hope that the other reviewers were more helpful with the results section of your paper. I hope my comments are helpful and I look forward to seeing future drafts or the final published version. Abstract This is appropriate as stands. If the authors have room it would be appropriate to add an implication to the conclusions section. Article Summary (Strengths and Limitations) This is appropriate as stands. Background The section unfortunately starts off weak. The authors start by saying that “the idea that patients and members of the public should be involved in health research has spread.” I know that the authors mean PPI, but the sentence comes across as research participation (e.g., qualitative interview participation). I am unsure what the authors mean by ‘spread.’ Spread where? Lay people will not understand what the WHO’s ‘Alma Alta Declaration’ is (line 7, page 5). The authors state on line 9, page 5 that PPI is done for normative or practical reasons. However, Sarah White has identified other reasons for PPI; nominal; instrumental; representative; and transformative. I provide a reference at the end of my review. The authors should consider citing Rosamund Snow, a patient-led researcher, who was highly critical of PPI. She argued that most of PPIE is performative, rather than allowing patients to have a real say in research.
--

The authors should consider the views of PPI members in their introduction. A paper by my colleagues Crocker et al (2016) found that PPI felt that PPI had a beneficial impact on health research.

The authors end with a justification for their study. However, I feel that it is not convincing enough. For example, they say “to date, research on PPI has been almost entirely qualitative” (line 33, page 5). This comes across as the authors opinion given that a quick PubMed search shows lots of quantitative articles on PPIE (particularly, RCTs). They then say, “we lack important information on how widespread PPI is within health research and in what circumstances researchers are more or less likely to include it.” We may lack this information but why is it important? What we gain from knowing it? This is unclear. Finally, they mention analysing the variation of reporting by location, method, topic, and funding source. I have a pretty good idea why this knowledge is important- but the authors should say why. Lay readers, of which many BMJ readers are, may not know why this information is important.

I was questioning why the authors sampled BMJ Open articles only, but there is good justification for this. Unfortunately, PPI reporting is not shown as a sign of quality in the consolidated reporting for qualitative research- something this article made me reflect on.

Methods

The methods is strong and well-written with sufficient information to understand how the study was conducted.

When the authors say ‘we’ it makes it seem like all authors did that part of the research method. It would be better to say ‘IL and AK read through each statement to check...’ for example. Unless the authors really do mean all authors.

It is unclear why the authors chose the year 2020 to identify research articles.

The ethics approval statement may be unnecessary since BMJ requires ethics statements as additional information.

If the authors are going to use subheadings in their methods, it is best to include one for ‘data collection/ extraction/ inclusion/ exclusion criteria’ (whichever works best for the authors).

Did the authors provide training to AK? They say that AK was involved in data analysis and interpretation but it unclear what her role was. I am glad to see that the authors listed to AK.

Results

Like I said previously in my review, I do not feel I am the right person to comment on the legitimacy of the results. What I will say is that the results read well, and the tables are superb. Unless I am missing something, I am a bit confused why there is a focus on China compared to say Europe which had a higher N and % in table 1.

Discussion

The discussion starts with a nice summary of the results. The authors could benefit from reflecting on why PPI may not be carried out (linking back to the nice introduction point about costs etc).

The authors claim that their results show what might increase PPI

usage. However, this unhelpful as they do not actually say what might increase PPI usage. The sentence that starts 'it seems clear that authors in some research fields are less persuaded of the importance of PPI.' This is true but the sentence makes no sense, especially the part where the authors say 'the researchers in those field might increase involvement.'

The authors rightly reflect on how the NIHR has set precedent for PPI usage. However, the authors fail to reflect on how the NIHR is one of the wealthier funders who can generally afford PPI. People who are funded by charities or smaller bodies tend not to have the funds for PPI. Moreover, people from certain countries may also be constricted by funding.

In the limitations paragraph the authors do tough on how the PPI statements may not reflect the experience of involvement. I would like to see the authors go the extra mile and say that they cannot say the included papers represent 'good PPI' (i.e., it is hard to say whether the PPI was performative or actually inclusive). Reporting is not enough- researchers need to be transparent about what PPI actually happened, rather than say 'PPI was included.'

The authors comment on cultural capital in their discussion. Lay readers may not understand what cultural capital is. There is an interesting paper by Locock et al on symbolic capital and PPI.

The comment about the GRIPP2 reporting checking list is an irrelevant point. We are told to follow reporting guidelines by journals but that doesn't we always cite them in the article.

Formatting/ Edits/ References

References appear to be correct, with the authors citing appropriate articles in good journals. The authors have done an excellent job of citing a diverse range of scholars.

There are some places where the authors could improve their grammar and punctuation.

1. First sentence in 'objectives' of abstract does not make sense.
2. Line 27, page 5, starting 'research on attitudes towards PPI' is a loaded sentence.
3. Full stop at the end of line 57, page 5, starting 'first author name.'
4. As above, line 5 and 9, page 6.
5. In the patient and public involvement statement, the authors do not need to say 'from the perspective of the professional co-authors' given that this is obvious.

Helpful References

White SC. Depoliticising development: the uses and abuses of participation. *Development in practice* 1996;6(1):6-15.

<https://blogs.bmj.com/bmj/2019/11/12/trisha-greenhalgh-towards-an-institute-for-patient-led-research/>

<https://blogs.bmj.com/bmj/2016/07/19/rosamund-snow-what-makes-a-real-patient/>

Crocker, J.C., Boylan, A.-M., Bostock, J. and Locock, L. (2017), Is it worth it? Patient and public views on the impact of their involvement

	in health research and its assessment: a UK-based qualitative interview study. Health Expect, 20: 519-528. https://doi.org/10.1111/hex.12479 Locock L, Boylan AM, Snow R, Staniszewska S. The power of symbolic capital in patient and public involvement in health research. Health Expect. 2017;20(5):836-844. doi:10.1111/hex.12519 Conclusion This article is interesting and sufficiently addresses it's aims. The methods are well-reported and the results are clear. However, there are several areas that need improving. For example, the rationale for the study is not convincing. The discussion also does not go far enough in regards to addressing the quality of PPI. I recommend that this article is accepted with major revision.
--	--

VERSION 1 – AUTHOR RESPONSE

RESPONSES TO REVIEWERS' COMMENTS FOLLOW

In the text below, the comments made by the reviewers are marked with ">" and our responses follow.

>Reviewer: 1

>Mr. Simon Collins, HIV i-Base

>Comments to the Author:

>Thank you for the chance to review this cross-sectional study that reports patient and public involvement (PPI) in roughly 20% of the research papers published in BMJ Open during 2020 (618/3000). Also for the regional breakdown by country and by type of study,

>The paper is well written and the results provide a snapshot indication that PPI is under used, with >less than half the studies from the UK (the country with highest report at 46%) reporting any >involvement.

>The results are important for showing how few studies currently use PPI globally. even in countries >where this concept is broadly understand, such as the UK.

>The following comments and mainly minor suggestions.

>Because the professional authors in the writing group are all from a single centre, >this perhaps misses an opportunity to include wider regional diversity, which might >have helped a study reporting on international differences.

We agree that greater diversity is a good thing, in most endeavours. The professional authors do all work in the same organisation and the most we can say is that they come from four different countries.

>Page 5 - lines 3-12

>How did the group recruit the individual PPI experience? Was this a friend or colleague >who also happened to have been a patient/carer?

AK is a member of the Peninsula Public Engagement Group (PenPEG) and expressed a particular interest in the topic of this paper, in which she already has experience. She has been a patient and carer for most of her life and has been active in both patient advocacy and PPI for over twenty-five years. We have changed the text to clarify AK's membership of this Group as well as of the extensive experience she brings.

>It would help to know why the group did not include someone who formally represented
>a patient or public group - and who had this broader experience.
AK was included on the basis of extensive work with patients and patient groups/organisations in her ten years of activity in the voluntary sector, which included crisis helpline volunteering, Trusteeships, being a senior post employee including five years as the CEO of a national charity. As noted above, we have changed the text to make clear AK's background and her membership of PenPEG.

>It is good that the PPI contribution is recognised at co-author level. However, this person
>was only engaged after the study had been designed. It would have made more sense for
>any study on PPI to really have started with PPI engagement from first principles.
We agree that such recognition is important. We can't do anything about this as far as this paper goes but our aim in future papers will be to ensure we have PPI engagement from as early as possible.

>This aspect of the study could perhaps be included as a study limitation. The groups own
>approach could perhaps be used in the discussion.
We were unsure what you meant when you wrote that we could use our own approach in the discussion but are happy to make relevant changes if you would like to clarify.

>General - Do the group think that there are studies where PPI engagement is not
>appropriate? It would be useful to give examples
We think that this is an important question but in this paper we wanted to assess the extent to which studies were reporting having done PPI. Discussions of whether PPI should or should not be part of specific studies would be both interesting and useful but we would have to expand this paper significantly in order to do that. Our current abstract is at the maximum permissible word limit! We do not have room to expand on this topic here but may well address it in future work, if other authors do not.

>Page 12 - line 10 onwards. This paragraph is sufficiently distinct from the first limitation
>to be referred to as a second limitation.
>The UK focus is a very important limitation given the international aspects of the findings.
We have changed the text to reflect this and we now describe three limitations instead of two.

>Perhaps the discussion could address likely large cultural and regional differences in the PPI
>concept, for example in general between high vs low income (in terms of level of support for
>research). Also, in UK/Canada/Australia and USA vs China/SE Asia etc (in terms of English
language).
>It might be helpful to include a table that summarises country/regional policies on PPI and the
>level of requirement in regional funding agencies.
Each of the things you mention is a very real issue (cultural differences, regional differences, national-income differences, linguistic differences) and worthy of further consideration. Our preference would be not to try to summarize PPI requirements on the part of funding bodies in different parts of the world – not only because we would need to find translators for each language who understands health research and PPI, but because even pinning down the policy for a single country and a single funder, such as the UK NIHR, can be different and there may be multiple relevant documents that may overlap, diverge, or even disagree. What we have done is extended the discussion on this topic and added some references that, we hope, pick up on at least some of the issues you raise.

>As this limitation could undermine the differences reported by region, the discussion could include
>a counter argument against this. Otherwise, the results currently read like a UK vs China
>comparison when the basis and understanding on PPI are likely to be very different in each setting
>(invalidating such a simple comparison).
We structured and presented our analyses so that we first showed the relationships with each of the independent variables (location, method, topic, and funder) and then, based on our adjusted regression models, the relationships with these variables when the other variables were adjusted for. The results shown in Figure 1 are of the relationship we found between PPI inclusion and location when funding sources was taken into account (as well as method and topic). This approach allows us to address the issue you mention, i.e., the potential that differences according to region might be undermined when funding source is accounted for. We recognise that adjustment of this type is a

broad-brush approach, all the more so when we are, as here, adjusting only for the larger funders. Nonetheless, the differences across locations appear robust to these adjustments: which is to say, there are still major differences between regions when we use regression models to test whether funding source makes any difference.

Emphasizing the UK-China differences is something we have done for rhetorical reasons rather than because of any particular interest in how these two locations differ. We focus on the difference between the UK and China because these are the places in which we found the highest and lowest levels of PPI inclusion. In terms of drawing attention to the variation in PPI inclusion that we found, contrasting the highest and lowest levels make the difference appear most stark. If the highest had been Germany and the lowest had been Central and South America then we would instead have focused on those locations in our reporting.

>Page 25, line 25. If point above is accepted, this should become the third limitation.
As noted above, we have changed the text to reflect those point and now note three limitations

>The use of one journal might not be a limitation, because it standardises one aspect of the
>study question.

At the same time, this does limit the generalisability of our results because one could say that what applies to the BMJ Open does not apply to other journals. In a way, this is both a strength and a weakness of our approach. We have added a sentence to the discussion to reflect the point you have raised.

>Perhaps comment in detail on why inclusion in countries with high recommendations to include
>PPI are still so limited.

On reflection we have decided that we would prefer not to do this. The work we report on in this paper gives us little insight into this matter and we feel we would be doing no more than speculating if we tried to address this issue.

>If the authors have limited information about policies in other regions and countries, this should be
>listed as a limitation. Otherwise, the paper is reporting on an issue that they know is established in
>their own country but commenting on research in very different settings.

We have added a sentence to (what is now) the second limitation that highlights the fact that we, as a group of authors, are familiar with UK policies and expectations around PPI but lack knowledge of how these things work in other parts of the world.

>Did the group consider the inclusion of a community rep as a co-author as an indication of the level
>of PPI - and for formally acknowledging this level of engagement?

We considered the community of experience in relation to this study to be people involved in research as patients or carers as part of PPI activities. AK is part of this community and had expressed an interest in being involved in PPI evaluations, hence her co-authorship of this paper. We hope that the changes we made in response to your previous comments, clarifying AK's background and role, are sufficient to address this point.

>Please could the discussion include suggestions for how the present level of PPI could be increased
> - especially when funders not only strongly encourage this but are willing to cover costs.

Again, we would prefer to avoid this because we can only speculate on why this might be. Further research is needed to answer this question.

>What about the role of publishers in influencing PPI (as a criterion for publication).

We can't comment on this here because we are dealing with papers published by a single publisher (BMJ) then we don't know what difference publisher policies and criteria make. As we have suggested, this is one reason why drawing on only a single journal represents a limitation for our study.

>Will future projects include contacting research studies without PPI to understand reasons for
>exclusion?

This is a very apt and important point. There is certainly potential for this, and it would be an interesting avenue to explore.

>Reviewer: 2
>Ms. Amelia Talbot, University of Oxford

>Comments to the Author:

>This research article presents a cross-sectional analysis of 3000 papers in the BMJ open published in 2020 to conclude on frequency of usage and reporting of patient and public engagement (PPI). I thank the BMJ Open and the authors for allowing me to review this insightful and generally well-written article which I hope will get funders, journals, and researcher to reflect on the importance of conducting and reporting PPI. This paper I feel is appropriately submitted to the BMJ Open given that it samples articles for this journal.

For full transparency, I am a qualitative researcher with limited experience in quantitative methods. So, my comments on your results are limited. I normally do not review non-qualitative papers. However, I am currently doing a massive research project on being a patient researcher (I research the condition I live with). As such, I have a vast array of knowledge on PPI and feel that my comments may be of use to you. I hope that the other reviewers were more helpful with the results section of your paper.

>I hope my comments are helpful and I look forward to seeing future drafts or the final published version.

>Abstract

>This is appropriate as stands. If the authors have room it would be appropriate to add an implication to the conclusions section.

We are, unfortunately, at the word limit already so can't add anything else.

>Article Summary (Strengths and Limitations)

>This is appropriate as stands.

>Background

>The section unfortunately starts off weak. The authors start by saying that "the idea that patients and members of the public should be involved in health research has spread." I know that the authors mean PPI, but the sentence comes across as research participation (e.g., qualitative interview participation). I am unsure what the authors mean by 'spread.' Spread where? We have rewritten the opening sentence to clarify both these points: what we mean by involvement, and what we mean by spread.

>Lay people will not understand what the WHO's 'Alma Alta Declaration' is (line 7, page 5). We have added a description of the Declaration.

>The authors state on line 9, page 5 that PPI is done for normative or practical reasons. However, Sarah White has identified other reasons for PPI; nominal; instrumental; representative; and transformative. I provide a reference at the end of my review. The authors should consider citing Rosamund Snow, a patient-led researcher, who was highly critical of PPI. She argued that most of PPIE is performative, rather than allowing patients to have a real say in research. These are crucially important issues in terms of how we think about and implement PPI. The work of both Sarah White and Rosamund Snow each constitutes to a significant counter-narrative about what and whom PPI is for. As members of a community of researchers who think about and "do" PPI we like to think that we are both aware of and sensitive to these issues of knowledge, representation, and power and so we entirely agree with you about the significance of these matters. Rather than squeeze the references you suggest into the introduction we have added them to the discussion section, where we think they fit well as an adjunct to changes we have made in relation to other of your comments.

In the introduction, we have amended the text to clarify that the normative vs practical aspects are something that those who argue for PPI use in making a case for it. Like you, we do not think that these things are the totality of what PPI actually is or does.

>The authors should consider the views of PPI members in their introduction. A paper by my colleagues Crocker et al (2016) found that PPI felt that PPI had a beneficial impact on health research.

Thank you. We have added this reference to the introduction.

>The authors end with a justification for their study. However, I feel that it is not convincing enough.
>For example, they say “to date, research on PPI has been almost entirely qualitative” (line 33, page >5). This comes across as the authors opinion given that a quick PubMed search shows lots of >quantitative articles on PPIE (particularly, RCTs). They then say, “we lack important information on >how widespread PPI is within health research and in what circumstances researchers are more or >less likely to include it.” We may lack this information but why is it important? What we gain from >knowing it? This is unclear. Finally, they mention analysing the variation of reporting by location, >method, topic, and funding source. I have a pretty good idea why this knowledge is important- but >the authors should say why. Lay readers, of which many BMJ readers are, may not know why this >information is important.

What we meant was that there were no quantitative studies on how widespread PPI inclusion in studies was. We have amended the text to clarify this. Our aim in this paper was, although we have not phrased it this way in the text, to carry out what we might call an epidemiology of PPI inclusion: that is, to use what information we could to figure out how common it was and in what groups it was more or less common. If our aim is to ensure that PPI is more constant part of health research – and this is certainly what is implied by the BMJ editorial policies on this issue as well as the actions of some funders – then it is hard to know (a) where we are and how successful activity to date has been, and (b) where success has been greatest and, conversely, where progress has been slow. We have amended the text to try to capture this idea, albeit in a much terser form.

>I was questioning why the authors sampled BMJ Open articles only, but there is good justification >for this. Unfortunately, PPI reporting is not shown as a sign of quality in the consolidated reporting >for qualitative research- something this article made me reflect on.

>Methods

>The methods is strong and well-written with sufficient information to understand how the study >was conducted.

>When the authors say ‘we’ it makes it seem like all authors did that part of the research method. It >would be better to say ‘IL and AK read through each statement to check...’ for example. Unless the >authors really do mean all authors.

Saying “we” is, of course, a shorthand. BMJ Open requires a Contributorship Statement and the guidance states: “Articles should list each author's contribution individually at the end”. We have done this so we do not think there is a need to incorporate this information in the body of the text. We are happy to be guided by the editor and to make this change if required.

>It is unclear why the authors chose the year 2020 to identify research articles.
We have added a statement explaining this.

>The ethics approval statement may be unnecessary since BMJ requires ethics statements as >additional information.

We have left the statement in for now but will remove it should the editors think this better.

>If the authors are going to use subheadings in their methods, it is best to include one for ‘data >collection/ extraction/ inclusion/ exclusion criteria’ (whichever works best for the authors).

We have added a subheading to cover this

>Did the authors provide training to AK? They say that AK was involved in data analysis and >interpretation but it unclear what her role was. I am glad to see that the authors listed to AK. AK has over twenty-five years research experience as a co-applicant and lay researcher, a contributor to papers, and a co-organiser of two conferences (one international). Her role in the production of this paper was as a member of the team who took part in data analysis, interpretation, and providing reflections from a non-academic perspective. She did not request any training in order to carry out these activities but training would have been provided had she done so. We have added a sentence to the Patient and Public Involvement section giving more detail about her background and experience.

>Results

>Like I said previously in my review, I do not feel I am the right person to comment on the legitimacy

>of the results. What I will say is that the results read well, and the tables are superb. Unless I am
>missing something, I am a bit confused why there is a focus on China compared to say Europe
>which had a higher N and % in table 1.

Thank you for your positive feedback. The focus on China was based not on the n and % in Table 1 but on the mean % of PPI inclusion in papers. We contrasted UK and China because the difference between them, in terms of PPI inclusion, was greatest and this provided the clearest indication of how wide the range in levels of inclusion was.

>Discussion

>The discussion starts with a nice summary of the results. The authors could benefit from reflecting
>on why PPI may not be carried out (linking back to the nice introduction point about costs etc). The
>authors claim that their results show what might increase PPI usage. However, this is unhelpful as
>they do not actually say what might increase PPI usage. The sentence that starts 'it seems clear
>that authors in some research fields are less persuaded of the importance of PPI.' This is true but
>the sentence makes no sense, especially the part where the authors say 'the researchers in those
>field might increase involvement.'

Your observation that we don't actually say what might increase PPI inclusion is entirely correct and, we now understand, this is because we were – in our previous wording – tending towards causal thinking in a situation where we could only make statements about correlation. We have rewritten the first part of this paragraph entirely and it now, we hope you will agree, gives a more accurate indication of what we can and cannot infer from our findings. Thank you for helping us clarify this.

>The authors rightly reflect on how the NIHR has set precedent for PPI usage. However, the authors
>fail to reflect on how the NIHR is one of the wealthier funders who can generally afford PPI. People
>who are funded by charities or smaller bodies tend not to have the funds for PPI. Moreover, people
>from certain countries may also be constricted by funding.

The NIHR is a large funder but paying for PPI inclusion represents an opportunity cost for funders of any size. If, for instance, paying PPI costs took up 1% of a funder's budget then the primary consequence of being a larger funder (presuming no economies of scale) would be that that 1% would amount to a larger total. In any given case, the decision to allocate a percentage of the research budget to PPI would mean that the funder could not use that funding in a different way. In addition, it is our experience that, at least in the UK, some charitable funding bodies (e.g., Stroke Association, Alzheimer's Society, The McPin Foundation) are extremely committed to PPI, are among the first funders to have supported PPI costs, and in some cases fund PPIE aspects over other research costs.

You are right that researchers in some parts of the world will not have access to funding to pay for PPI. This is borne out in our findings about the variation of PPI in studies from different locations.

>In the limitations paragraph the authors do touch on how the PPI statements may not reflect the
>experience of involvement. I would like to see the authors go the extra mile and say that they
>cannot say the included papers represent 'good PPI' (i.e., it is hard to say whether the PPI was
>performative or actually inclusive). Reporting is not enough- researchers need to be transparent
>about what PPI actually happened, rather than say 'PPI was included.'

We are more than happy to go this extra mile and clarify that we can't say anything about the quality of the PPI included. We have not used the phrase "good PPI" because we think that would open up a much bigger issue concerning what "good" means that we lack space to develop in this paper. We have also included the references to the work of White and Snow that you suggested.

>The authors comment on cultural capital in their discussion. Lay readers may not understand what
>cultural capital is. There is an interesting paper by Locock et al on symbolic capital and PPI.
Thank you for pointing this out. We have replaced this technical term with the phrase "cultural resources", which is what the author of the text we draw on uses in their own lay summary.

>The comment about the GRIPP2 reporting checking list is an irrelevant point. We are told to follow
>reporting guidelines by journals but that doesn't we always cite them in the article.
Our intention here was to link back to the idea of "good PPI", based on the premise that reporting of PPI using criteria like GRIPP2 would be of higher quality. However, in light of your comment we have remove this point.

>Formatting/ Edits/ References

>References appear to be correct, with the authors citing appropriate articles in good journals. The
 >authors have done an excellent job of citing a diverse range of scholars.
 >There are some places where the authors could improve their grammar and punctuation.
 >1. First sentence in 'objectives' of abstract does not make sense.
 >2. Line 27, page 5, starting 'research on attitudes towards PPI' is a loaded sentence.
 >3. Full stop at the end of line 57, page 5, starting 'first author name.'
 >4. As above, line 5 and 9, page 6.
 Thank you for pointing out these errors. We have corrected them.

>5. In the patient and public involvement statement, the authors do not need to say 'from the
 >perspective of the professional co-authors' given that this is obvious.
 We have not changed this one. If we do not say "from the perspective of the professional co-authors"
 then the impression given would be that the authorial voice is that of those professional co-authors. It
 is not: the authorial voice is that of the professional and non-professional authors together, and this is
 the "we" who speaks in this paper. We have left this phrase as it is in order to make clear that, in this
 case alone, we are separating out the "professional co-author" voice in order to comment on AK's
 contributions.

>Helpful References

>White SC. Depoliticising development: the uses and abuses of participation. Development in
 >practice 1996;6(1):6-15.

><https://blogs.bmj.com/bmj/2019/11/12/trisha-greenhalgh-towards-an-institute-for-patient-led-research/>

><https://blogs.bmj.com/bmj/2016/07/19/rosamund-snow-what-makes-a-real-patient/>

>Crocker, J.C., Boylan, A.-M., Bostock, J. and Locock, L. (2017), Is it worth it? Patient and public
 views

>on the impact of their involvement in health research and its assessment: a UK-based qualitative
 >interview study. Health Expect, 20: 519-528.

><https://www.ncbi.nlm.nih.gov/pmc/articles/PMC5433537/>

>Locock L, Boylan AM, Snow R, Staniszewska S. The power of symbolic capital in patient and public
 >involvement in health research. Health Expect. 2017;20(5):836-844. doi:10.1111/hex.12519

>Conclusion

>This article is interesting and sufficiently addresses it's aims. The methods are well-reported and
 >the results are clear. However, there are several areas that need improving. For example, the
 >rationale for the study is not convincing. The discussion also does not go far enough in regards to
 >addressing the quality of PPI.

>I recommend that this article is accepted with major revision.

VERSION 2 – REVIEW

REVIEWER	Collins, Simon HIV i-Base
REVIEW RETURNED	05-May-2022

GENERAL COMMENTS	Thank you for the time and consideration taken over the review comments. While I would have welcomed some of the discussions that I tried to draw you out on, I appreciate the changes that have been made. Good luck with your future research, where some of this might be considered :)
--

REVIEWER	Talbot, Amelia
-----------------	----------------

	University of Oxford, Nuffield Department of Primary Health Care Sciences
REVIEW RETURNED	05-May-2022

GENERAL COMMENTS	Thank you for your kind responses to my review on your manuscript. You have fairly addressed all of my comments and I am confident that this manuscript is a publishable standard. Well done on publishing such a fantastic and important manuscript. I hope to read more work by you in the future.
--